# Effect of Orange Juice on the Properties of Heat-Polymerized and 3D-Printed Denture Materials

**DOI:** 10.3390/polym17010056

**Published:** 2024-12-29

**Authors:** Büşra Tosun, Zeynep Öztürk, Nur Uysal

**Affiliations:** 1Department of Prosthodontics, Faculty of Dentistry, University of Abant İzzet Baysal, 14030 Bolu, Turkey; nur.uysal@ibu.edu.tr; 2Department of Pedodontics, Faculty of Dentistry, University of Abant İzzet Baysal, 14030 Bolu, Turkey; zeynep.ozturk@ibu.edu.tr

**Keywords:** heat-polymerized PMMA, denture base, microhardness, natural, optical, roughness, 3D printing

## Abstract

This study evaluated the color stability, surface roughness, and hardness of 3D-printed and heat-polymerized denture materials. A total of 90 samples were prepared, with equal numbers of 3D-printed and heat-polymerized disks. The initial hardness, surface roughness, and color values of the samples were measured. After 14 days of immersion in distilled water, natural orange juice, or commercial orange juice, the measurements were repeated. Based on the findings, 3D-printed samples exhibited a greater reduction in Vickers hardness (56.24 ± 15.81%) compared to heat-polymerized samples (18.93 ± 11.41%). Materials immersed in commercial orange juice exhibited a greater reduction in hardness compared to those in other solutions (43.13 ± 23.66). Surface roughness increased by 46.66 ± 26.8% in heat-polymerized samples and by 26.16 ± 20.78% in 3D-printed samples, with the highest increase observed in commercial orange juice (50.73 ± 28.8%) (*p* < 0.001). The color change (ΔE) was significantly higher in heat-polymerized samples (ΔE = 5.05 ± 0.28) than in 3D-printed samples (ΔE = 3.9 ± 0.26) (*p* < 0.001). This study demonstrates that the material type and immersion solutions play a critical role in determining the mechanical and optical properties of denture materials, with commercial orange juice having the most pronounced effect on surface roughness and hardness.

## 1. Introduction

Partial and complete edentulism are commonly managed using removable prostheses, which are frequently preferred treatment options for patients requiring prosthetic rehabilitation. Among the materials used in the fabrication of these prostheses, polymethyl methacrylate (PMMA) resins are considered versatile due to their ability to be polymerized through various methods, including heat, chemical, microwave, or light activation. Heat-polymerized PMMA is particularly favored in the fabrication of complete dentures due to its low cost, ease of use and handling, and its ability to provide satisfactory physical and mechanical properties along with adequate esthetics. These attributes make it a widely adopted choice in dental practice for producing durable and esthetically pleasing dentures [1,2].

A denture base material should possess a smooth and glass-like surface, which ensures a harmonious integration with the natural appearance of soft tissues to provide esthetic continuity. It is crucial to maintain both color stability and translucency throughout the manufacturing process, as these materials are expected to resist staining and color changes during clinical use. However, color changes in acrylic resins can lead to significant esthetic issues, potentially affecting patient satisfaction and the long-term appearance of the prostheses [3]. Excessive heat generation or insufficient pressure during the polymerization process can result in the formation of residual monomer content and a porous structure in the material. These issues may compromise the mechanical integrity and durability of the denture base, potentially leading to reduced clinical performance and increased susceptibility to discoloration and microbial colonization [4,5]. Such porosities can make the denture surface more susceptible to color changes, particularly after exposure to staining beverages such as coffee, tea, and cola [6]. Moreover, the acid content of these beverages is also a critical factor that affects the color stability of denture base materials [7,8]. Color change is considered an indicator of the aging of the prosthetic material or damage resulting from environmental conditions [9].

Fruit juices contain significant amounts of vitamins, minerals, antioxidants, and phytochemicals. Thus, they not only boost the immune system but also fulfill daily nutritional requirements [10]. However, a single serving of sweetened fruit juice beverages contains more sugar than the recommended daily intake for an adult. Consequently, their health benefits are lower compared to those of natural fruit juices [11]. In addition to their medical effects, it is also crucial to understand the impact of these beverages on dental materials. The rate of color change in restorations can be influenced by several factors, including water absorption, chemical interactions, dietary habits, smoking, poor oral hygiene, and surface smoothness [12,13]. Moreover, the chemical components in certain beverages can weaken the surface structure of restorations, thereby leading to degradation and resulting in unsightly external pigmentation [14].

The surface roughness of denture base materials is critical for maintaining the health of oral tissues, given that the prosthesis is in direct contact with these tissues. Microorganisms present in the oral environment can more easily adhere to rough surfaces, forming colonies that can increase the risk of infections [15]. The literature reports a direct correlation between surface roughness and the accumulation of dental plaque as well as the colonization of Candida albicans [16,17]. This can complicate oral hygiene for denture wearers, potentially leading to oral health issues. In addition, hardness is a critical mechanical property that determines the durability of denture base materials. Denture bases with higher hardness levels facilitate finishing processes and provide resistance to wear during cleaning [18,19,20]. Therefore, the hardness of dentures is considered a critical parameter for both clinical use and long-term durability.

In recent years, the rapid digitalization in dentistry has opened the door to innovative approaches in dental practice. The widespread adoption of 3D printing technology is particularly regarded as a revolutionary advancement in the design and production of dentures [21,22]. This technology has reduced the workload of dentists and dental technicians by enabling the production of dentures with higher precision within a shorter timeframe [23,24]. Additionally, the advantages of dentures produced with 3D printers include shorter production times, cost-effectiveness, fewer patient appointments, and improved patient comfort [23,25].

In dentistry, the most frequently preferred 3D printers are stereolithography (SLA) and digital light processing (DLP) models, depending on the applied method. In these devices, the printing platform is immersed in resin cured by ultraviolet (UV) or light-emitting diode (LED) light. The laser or LED light creates and shapes each layer of the object, repeating the process until the final print is completed [26,27,28]. While numerous studies have been conducted on CAD-CAM milled blocks [29,30,31], research on 3D-printed resins using DLP technology remains limited. With the increasing popularity of additive manufacturing techniques such as 3D printing, research in this field is becoming increasingly significant. However, to fully establish the clinical efficacy of this new technology, further research is needed, particularly regarding its mechanical, physical, and esthetic properties. Given the frequent consumption of fruit juices, this study aims to compare the color stability, surface roughness, and hardness properties of PMMA resins produced using 3D printing and conventional heat polymerization in beverages with varying acidity levels and chemical compositions (distilled water, commercial orange juice, natural orange juice). Thus, the null hypothesis of the present study is that natural and commercially produced orange juice does not have a significant impact on the optical and mechanical properties of denture base materials produced using either 3D printing or conventional methods.

## 2. Materials and Methods

### 2.1. Power Analysis and Sample Size

A power analysis was conducted using G*Power software (version 3.0.10; Heinrich Heine University, Düsseldorf), which determined that a minimum of 15 samples per group was required to achieve 80% power (α = 0.05).

### 2.2. Sample Design and Preparation

Ninety disk-shaped samples were prepared and divided into two groups. The first group consisted of 45 samples produced using a 3D printer, while the second group included 45 samples fabricated using conventional heat polymerization of acrylic resin (10 mm diameter × 2 mm thickness). All samples were digitally designed using design software (Meshmixer; Autodesk Inc., San Rafael, CA, USA).

The first group of samples was produced using a DentaFab 3D printer with PowerResins Denture (PR) resin (3BFAB Technology Inc., Istanbul, Turkey). All printing parameters were standardized to ensure that each sample was manufactured under identical conditions. After printing, the samples were cleaned in a washing unit containing 96% ethanol (Twin Tornado, Medifive, Seoul, South Korea) according to the manufacturer’s instructions and then dried using compressed air. After the cleaning process, the samples were cured in a curing unit (Medifive Twin Cure, Seoul, South Korea) for 30 min.

The conventionally produced PMMA samples were prepared using the flasking technique and shaped via the lost wax method. After polymerization, the samples were carefully trimmed and meticulously polished. The polishing process was conducted in three stages (coarse, medium, and fine grit) using polishing wheels at 300 rpm for 15 s per stage by a single operator. The thickness of each sample was measured using a digital caliper (Digimatic AOS, Mitutoyo AG, Tokyo, Japan) with an accuracy of 0.001 mm. The figures related to the 3D printing and conventional production processes are presented in Figure 1a,b.

The 90 samples from both groups were randomly divided into 3 subgroups, with 15 samples in each, designed to be immersed in different solutions. The experimental groups were defined in the following manner:Group 1: Samples immersed in distilled water.Group 2: Samples immersed in commercial orange juice (Cappy, Turkey).Group 3: Samples immersed in natural orange juice.

The pH values and compositions of the solutions are presented in Table 1.

### 2.3. Measurement of Microhardness

The surface microhardness of all samples was tested using a microhardness tester (Shimadzu HMV-2000, Kyoto, Japan). The device’s pyramid-shaped indenter was applied to the surface with a force of 100× *g* for 15 s. The surface of each sample was measured three times, and the average hardness value was calculated. On the 14th day, the samples were removed from the solutions and were rinsed with distilled water; thereafter, the microhardness measurements were repeated. The percentage reduction in hardness was calculated using the following formula:Percentage reduction = [(Initial hardness − Final hardness)/Initial hardness] × 100.

This formula was used to determine the relative reduction between the initial and final hardness values.

The experimental procedure for microhardness measurement is presented in Figure 2.

### 2.4. Measurement of Surface Roughness

The initial surface roughness of the samples was measured using a surface profilometer (HandySurf, Accretech, Japan) The profilometer was standardized with a cut-off speed of 0.25 mm, a reading length of 1.25 mm, and a measurement speed of 0.05 mm/s. Each sample’s surface was measured at three different points, and the average surface roughness values (Ra) were recorded. On the 14th day, after the samples were removed from the solutions and rinsed with distilled water, surface roughness measurements were repeated. The percentage increase in surface roughness was calculated using the following formula:Percentage increase = [(Final roughness − Initial roughness)/Initial roughness] × 100.

This formula was used to determine the relative increase in surface roughness between initial and final measurements.

The experimental procedure for surface roughness measurement is presented in Figure 3.

### 2.5. Color Evaluation

Before immersion in the solutions, the initial L, a, and b* color values** of the samples were measured using a spectrophotometer (VITA Easyshade^®^ V, Zahnfabrik, Germany). After being immersed in the designated solutions for specified durations, the samples were rinsed with distilled water, and the L, a, and b* values were remeasured on days 1, 3, 7, and 14. As stated in Güler et al. [32], a 24 h immersion period simulates approximately 1 month of beverage consumption. Based on this approach, 14 days was chosen as the maximum simulated exposure period in the present study and it was assumed that this corresponds to a period of approximately 14 months in real life conditions. To evaluate the color changes (ΔE), the values were calculated using the formula below [33]:∆E = [(L_2_ − L_1_)^2^ + (a_2_ − a_1_)^2^ +(b_2_ − b_1_)^2^]^½^.

### 2.6. Statistical Analysis

The data were analyzed using Minitab v14 and Jamovi v2.3.28. Normality was assessed using the Shapiro–Wilk test and skewness–kurtosis values. For data that followed a normal distribution based on material and solution, generalized linear models and multiple comparisons using Tukey HSD tests were employed. For data that did not follow a normal distribution based on material, solution, and time, a robust ANOVA was conducted using the Walrus package, with multiple comparisons analyzed using the Bonferroni test. Quantitative variables were presented as mean ± standard deviation for normally distributed data, while trimmed mean ± standard error values were used for non-normally distributed data. The level of significance was set at *p* < 0.05.

The experimental procedure for color evaluation is presented in Figure 4.

## 3. Results

### 3.1. Microhardness

The main effect of the material, excluding the solution factor, was found to be statistically significant for the decrease in Vickers hardness (%) (*p* < 0.001). The mean decrease in Vickers hardness (%) was 56.24 for the 3D-printed material, whereas it was 18.93 for the conventional PMMA. The main effect of the solution, excluding the material factor, was also statistically significant for the decrease in Vickers hardness (%) (*p* = 0.029). The highest mean decrease in Vickers hardness (%) was observed in commercial orange juice, while the lowest mean decrease was found in distilled water. Further, the mean decrease in Vickers hardness (%) differed between distilled water and commercial orange juice, while the values for natural orange juice were similar to those in the other two solutions. The interaction between material and solution was not statistically significant for the decrease in Vickers hardness (%) (*p* = 0.492). A comparison of the decrease in Vickers hardness (%) based on material and solution is presented in Table 2, with descriptive statistics in Table 3. The mean decrease in Vickers hardness (%) according to the material and solution is illustrated in Figure 5.

### 3.2. Roughness

The main effect of the material, excluding the solution factor, was found to be statistically significant for the increase in surface roughness (%) (*p* < 0.001). The mean increase in surface roughness (%) was 26.16 for the 3D-printed material, while it was 46.66 for the conventional PMMA. The effect of the solution, excluding the material factor, was also statistically significant for the increase in surface roughness (%) (*p* = 0.002). The highest mean value was observed in commercial orange juice (50.73%), while the lowest was in distilled water (26.13%). The mean values of the increase in surface roughness (%) were similar in distilled water and natural orange juice, while the values for commercial orange juice differed from those for the other two solutions. The interaction between material and solution was not statistically significant for the increase in surface roughness (%) (*p* = 0.083).

The highest increase in surface roughness (%) was observed in the interaction between conventional PMMA and commercial orange juice (69.83%), while the lowest increase was seen in the interaction between 3D-printed material and distilled water (20.96%). A comparison of the increase in surface roughness (%) based on material and solution is presented in Table 4, with descriptive statistics in Table 5. In addition, the mean increase in surface roughness (%) according to material and solution is illustrated in Figure 6.

### 3.3. Evaluation of Color Change

The effect of the material factor color change (ΔE) values, excluding the solution and time factors, was found to be statistically significant (*p* = 0.004). The mean ΔE value for the 3D-printed material was 3.9, which was lower than that of the conventional PMMA material (5.05). Additionally, the effect of the solution factor on ΔE values, excluding the material and time factors, was statistically significant (*p* = 0.002). The mean ΔE values were 5.79 for distilled water, 4.16 for commercial orange juice, and 3.62 for natural orange juice. Thus, it is evident that the ΔE values for commercial and natural orange juice were similar, whereas the values for distilled water differed from those of the other two solutions.

The highest ΔE value for the material and solution interaction was observed in the conventional PMMA group immersed in distilled water (6.09), while the lowest was in the 3D-printed PMMA group immersed in natural orange juice (2.96). In terms of the solution and time interaction, the highest ΔE value was recorded on day 14 in the distilled water group, while the lowest was observed on day 1 in the natural orange juice group. A comparison of ΔE values based on material, solution, and time is presented in Table 6, with descriptive statistics in Table 7. In addition, the mean ΔE values according to material, solution, and time are illustrated in Figure 7.

## 4. Discussion

In this study, the color stability, surface roughness, and hardness properties of PMMA resins produced using 3D printing and conventional heat polymerization were evaluated and compared. The samples were exposed to three different solutions (distilled water, commercial orange juice, and natural orange juice), and measurements were repeated at specific intervals. According to the findings, natural and commercial orange juice exhibited different effects on the optical and mechanical properties of the various denture base materials, thus leading to the rejection of the null hypothesis.

Hardness is one of the most crucial mechanical properties of denture base materials, and is closely related to their resistance to intraoral pressures and wear [34]. The hardness value reflects a material’s resistance to local plastic deformation and can significantly impact the longevity of the prosthesis [35,36]. Prostheses made from materials with low surface hardness may be damaged during mechanical brushing, thereby leading to increased plaque accumulation, discoloration, and deterioration of the prosthesis’s esthetic appearance [35,37]. Consequently, the survival rate of the prosthesis may be significantly reduced.

Al-Dwairi et al. [36] reported that the comparison of the mechanical properties of denture base resins produced with 3D printing and conventional PMMA in their study revealed that the heat-polymerized conventional PMMA group achieved the highest hardness and surface roughness values. However, in the present study, it was found that while the initial hardness of 3D-printed PMMA denture base materials was higher than that of conventionally heat-polymerized PMMA, the former exhibited more pronounced hardness loss after exposure to different solutions. These findings can be attributed to the internal structural differences in materials produced using photopolymerization technology. A study by Revilla-León and Özcan [38] emphasized that materials manufactured using photopolymerization in 3D printing technologies achieved rapid surface hardening; however, due to incomplete polymerization in the inner layers, a high amount of residual monomer may remain. The presence of elevated levels of residual monomer can negatively affect properties such as the water absorption and solubility of denture base materials. This, in turn, may lead to a greater loss of hardness when exposed to acidic solutions, thereby adversely affecting the mechanical strength of photopolymerized materials [38].

The layer-by-layer manufacturing process of 3D-printed materials and the lower double-bond conversion rate can lead to insufficient curing of each layer. This can negatively affect the mechanical properties of the material by reducing chain crosslinking efficiency [39]. In agreement with our current findings, Prypic et al. [35] reported that 3D-printed denture base materials had lower surface hardness values compared to other acrylic materials tested, including CAD/CAM and conventional heat-polymerized resins. The decrease in hardness can be attributed to factors such as the composition of the material, stress layers, thermal stress, and water absorption [35].

In addition, it has been reported in the literature that the mechanical properties of PMMA prosthetic base materials are affected by factors such as polymerization method, degree of crosslinking, and chemical composition [40]. In particular, the curing time plays a critical role in the hardness values of 3D-printed resins. Studies have shown that extending the curing time significantly increases the crosslinking rate (degree of conversion, DC), leading to higher hardness values [40]. In the present study, a curing time of 30 min was used in accordance with the manufacturer’s instructions. It is thought that if the curing time is extended, the hardness loss in 3D-printed samples may be less and the results may differ. However, although the 3D printing process offers the advantage of fast production, the layer-by-layer manufacturing mechanism may lead to insufficient curing of each layer. This can lead to insufficient curing density throughout the structure and a significant reduction in chain crosslinking efficiency [41]. These limitations in the printing process may explain the observed differences in mechanical properties between 3D-printed and conventional denture base materials.

Further, a smooth denture surface helps reduce plaque and microbial accumulation, as microbial adhesion and colonization typically occur on rough surfaces [42]. A rough surface also makes the material more susceptible to staining, which can lead to unpleasant odors [43,44]. Several studies that have evaluated the mechanical and surface properties of denture base materials produced using 3D printing resins and conventional PMMA resins have reported that the groups made with conventional resin exhibit higher surface roughness values [36,45]. Similarly, in our study, it was found that denture base materials produced using conventional methods exhibited higher surface roughness compared to those produced with 3D printing. This can be attributed to the variations in heat and pressure used in traditional polymerization processes, which may cause irregularities and microvoids among polymer chains. Similar to the findings of our study, Di Fiore et al. reported that, among milled, 3D-printed, and heat-polymerized PMMA resins for denture bases, the 3D-printed PMMA resin group showed smoother surface properties than the conventional PMMA group [46].

The traditional prosthetic base fabrication method has been widely used for many years; however, problems such as porosity, surface roughness, and volumetric/linear shrinkage have been frequently reported [47,48]. It has also been reported that rough surfaces retain more microorganisms than smooth surfaces and create a protected environment that is difficult to remove with oral hygiene methods [49,50].

Prolonged contact of resin-based materials with environments containing large amounts of water and acids can lead to expansion of the resin matrix, hydrolysis of the silane, and consequently the formation of microcracks and voids [8]. In the literature, it has been reported that beverages containing acidity regulators such as phosphoric acid, malic acid and citric acid cause hydrolysis of ester radicals in PMMA monomers and disruption of the structural integrity of resin composite materials [51,52]. Citric acid and sodium citrate in beverages such as commercial orange juice are examples of these effects. In particular, citric acid, due to its high calcium chelating capacity, can remove calcium from the saliva or prosthesis surface, making the resin surface more sensitive. Therefore, beverages with low pH and citric acid are known to have the highest corrosive effect on PMMA materials [53]. In this study, the highest increase in surface roughness (%) was observed in samples immersed in commercial orange juice, while the lowest increase was detected in the distilled water group. Our findings are in agreement with the study conducted by Al Taweel et al. [43]. In that study, it was revealed that acidic beverages with low pH values, such as Coca-Cola and lemon juice, caused a significant increase in the surface roughness (Ra) of PMMA prosthetic resins. This effect was attributed to the fact that acidic components such as phosphoric acid and citric acid disrupt the structural integrity of the polymer surface, leading to dissolution and softening [54]. Again, Abu-Bakr et al. [55] revealed that low pH environments (such as orange juice and Coca-Cola) negatively affect the color stability and surface integrity of compomers. It was reported that under acidic conditions, the surface of the copolymer softened significantly due to ion loss from the glass phase, resulting in a rougher structure with voids on the surface. The study showed that this could be attributed to the separation of individual particles [55].

It has been shown that beverages with low pH levels—such as coffee, tea, lemon juice, orange juice, red wine, and cola—as well as the pigments they contain have detrimental effects on the surface of acrylic resin dentures, thereby leading to an increase in surface roughness [38]. This can be explained by changes in the properties of the polymerized materials as well as the potential infiltration of certain components present in commercial orange juice, which may contribute to surface abrasion [42].

Increased acidity in the oral environment can lead to many adverse conditions, such as dental caries, periodontal disease, taste disturbance, bad breath, and oral infections. Patients using dental prostheses may experience denture stomatitis as a result of fungal growth due to low pH. Furthermore, the effects of a decrease in oral pH on dental prosthetic acrylic resins are significant. Continuous contact with acidic saliva can lead to degradation of the acrylic material, a decrease in microhardness, and the release of more residual monomers. This can lead to a decrease in the fracture resistance of the material and make prostheses more brittle [56,57]. In a study conducted by Poggio et al. [58], it was reported that microhardness of resin composites decreased when immersed in acidic solutions, but no significant change in microhardness was observed when immersed in distilled water. Similarly, Tinaztepe et al. [59] reported that higher roughness and hardness changes were observed in the groups immersed in low pH environment in their study investigating the hardness and surface roughness of prosthesis base acrylic resins produced by different methods. In our current study, the highest Vickers hardness loss (%) was observed in commercial orange juice with low pH, while the lowest hardness loss was recorded in distilled water.

In the literature, it has been reported that natural orange juice has higher sodium, potassium, calcium, and copper content compared to commercial orange juice [60]. The higher mineral concentrations contained in natural orange juice are thought to cause less dissolution or degradation of the PMMA surface. In contrast, it is suggested that the lower mineral content and acidic modifiers contained in commercial orange juice may have led to a greater loss of surface hardness and more pronounced increases in surface roughness.

Spectrophotometry is one of the most recommended and accurate methods for objectively and quantitatively evaluating the color change of a material [61]. This method provides the ΔE (ΔEab) value, which represents color changes after various processes by calculating the L*, a*, and b* values [61]. In this study, the color change of denture base materials was recorded using a spectrophotometer in terms of L*, a*, and b* values.

The findings revealed that the mean ΔE value for conventionally produced PMMA was statistically significantly higher than that of the 3D-printed material (*p* = 0.004). These results are consistent with a similar study conducted by Alfouzan et al., in which 3D-printed and conventional PMMA samples were immersed in coffee, cola, lemon juice, and artificial saliva. It was found that the color change was greater in the conventionally produced samples compared to the 3D-printed ones [62]. The differences in ΔE values for color change are believed to be related to alterations in the optical properties of resin materials resulting from water absorption [63]. Resin materials that can absorb water are also prone to absorbing other liquids that may cause color deterioration [8]. This can be explained by the surface structure and polymerization processes of 3D-printed materials, which result in lower water absorption and, consequently, greater color stability compared to conventionally produced materials.

In the present study, the highest mean ΔE value was observed in distilled water, followed by those in commercial orange juice and natural orange juice. The staining potential of liquids varies based on their composition, pH levels, and other characteristics [8]. In the study conducted by Bezgin et al. [64], distilled water—which was selected as the negative control group—produced perceptible color differences detectable by the human eye in the measurements taken on days 1 and 60. In studies conducted to evaluate the effect of distilled water on the color change of restorative materials [5,65,66,67], statistically significant differences were found between the initial and post-immersion color values. Although distilled water has a neutral pH, it can still cause certain changes in the polymer structure due to water absorption. When PMMA interacts with water, it can absorb moisture, causing its structure to degrade [51]. This degradation may break the material’s polymer chains, resulting in surface dullness and noticeable changes in color. The changes after prolonged immersion can be attributed to the hygroscopic absorption of water within the material [5,63,67]. Buchalla et al. [63] also indicated that changes in the optical properties of materials due to water absorption could be responsible for the variations in ΔE values.

Some authors have reported that the water content of mouthwashes may affect discoloration; therefore, the effects seen in mouthwashes may also be observed in the distilled water (control) group [68,69]. Ulusoy et al. also observed discoloration in all samples stored in distilled water in their study [70]. In accordance with these studies, color changes were observed in the distilled water groups in the present study. The water absorption of resin materials is primarily affected by their chemical composition and the cavities they contain [71]. The chemical composition plays an important role in the chemical interaction with water molecules; this is particularly related to the hydrophilic polar regions in residual monomers, which tend to interact with water molecules [71,72]. While the coloration effect of distilled water was expected to be the lowest among the groups, it is thought that the opposite result obtained may be due to the distillation method and device used [69].

The most abundant organic acids in orange juices are citric and malic acids. Additionally, orange juices are good sources of ascorbic acid. However, commercial orange juices contain lower amounts of ascorbic acid, malic acid, and oxalic acid compared to natural orange juices [73]. The differences in the ΔE values of commercial and natural orange juices are believed to be due to their varying organic acid contents as well as the presence of colorants, flavoring agents, and other artificial additives in commercial orange juice. In a study conducted by Savas et al. [74], which examined the color changes of restorative materials immersed in different solutions, it was found that mildly acidic chocolate milk (pH: 6.4) produced the greatest color change, while highly acidic orange juice (pH: 3.6) resulted in the lowest color change. Similarly, Bagheri et al. [8] found that despite its low pH, cola exhibited a lower staining capacity compared to other substances (red wine, coffee, tea, soy sauce). Therefore, it was concluded that the color changes observed after immersion in various solutions cannot be attributed solely to surface changes related to pH levels [74]. Consistent with these findings, the highest ΔE value for the interaction between material and solution in the present study was observed in the conventionally produced PMMA group immersed in distilled water, while the lowest ΔE value was found in the 3D-printed group immersed in natural orange juice.

These findings highlight important considerations for clinical decisions in the choice of prosthetic base material. The higher loss of microhardness observed with 3D-printed materials may warrant caution in patients with high consumption of acidic beverages. However, the lower surface roughness and discoloration of these materials make them a more preferable option for patients who prioritize esthetics and oral hygiene. Conventional PMMA materials, despite exhibiting higher hardness stability, may require more frequent maintenance due to their higher surface roughness and tendency toward stain retention. Our findings also reveal that commercial orange juice shows more negative effects on material properties compared to natural orange juice. Commercial orange juice caused a greater loss of hardness and more pronounced increases in surface roughness. This supports the hypothesis that the acidic modifiers and lower mineral content in commercial orange juice accelerate degradation of the PMMA surface. In contrast, the higher mineral concentrations of natural orange juice may have shown a relatively protective effect, leading to less surface dissolution and degradation.

These results emphasize the need to consider not only the choice of material but also the dietary habits of prosthesis wearers. In particular, regular consumption of commercial acidic beverages may adversely affect the mechanical and esthetic properties of prosthetic materials, and it is of great importance for clinicians to provide appropriate care recommendations to patients. Material selection should be tailored to the individual needs and lifestyles of patients, balancing durability with esthetic and functional requirements.

The present study has several limitations. As this study could not fully replicate intraoral conditions, these in vitro results need to be further validated under in vivo conditions. Nevertheless, the current findings can provide valuable guidance for clinical applications. No aging process was applied to the samples in this study. Additionally, the oral environment is influenced by various factors—such as dietary habits, hygiene practices, and saliva—which could also have impacted the observed results. Furthermore, the immersion durations used in this study were limited to simulate short- to mid-term exposures. Longer immersion periods are needed to better represent long-term clinical conditions and provide a more comprehensive understanding of the tested solutions’ effects on denture base materials.

## 5. Conclusions

Given the limitations of this study, the following conclusions can be drawn:The reduction in microhardness of the denture base material produced with 3D printer resin was greater than that for the material produced with conventional PMMA. There was no significant difference in the reduction in hardness between commercial and natural orange juice.The increase in surface roughness was higher in the denture base material produced with conventional PMMA compared to the 3D-printed group. Commercial orange juice significantly increased the surface roughness of materials compared to natural orange juice.The 3D-printed material exhibited lower color change compared to the material produced with conventional PMMA.Distilled water showed the highest ΔE value among all materials, while no significant difference in color change was observed between the natural and commercial orange juice groups.

## Figures and Tables

**Figure 1 polymers-17-00056-f001:**
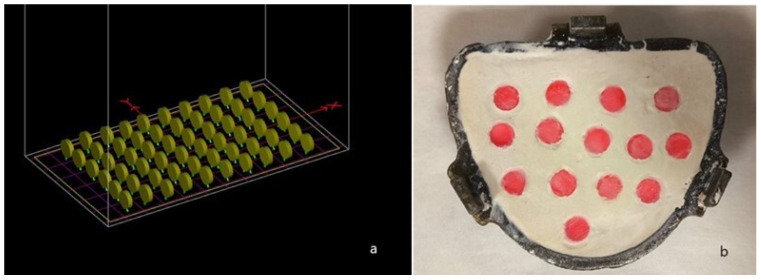
Specimens placed in the slicing software for 3D printing (**a**); specimens prepared in the flask for production using the conventional method (**b**).

**Figure 2 polymers-17-00056-f002:**
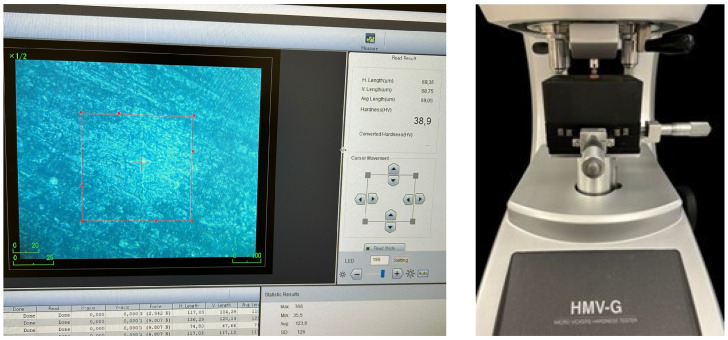
The experimental procedure for microhardness measurement.

**Figure 3 polymers-17-00056-f003:**
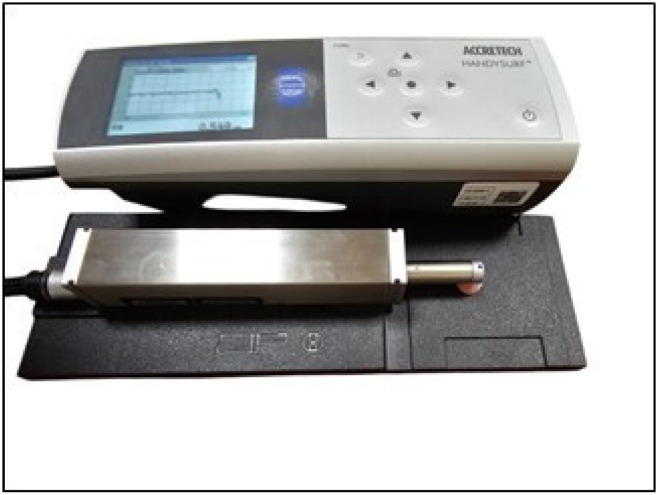
The experimental procedure for surface roughness measurement.

**Figure 4 polymers-17-00056-f004:**
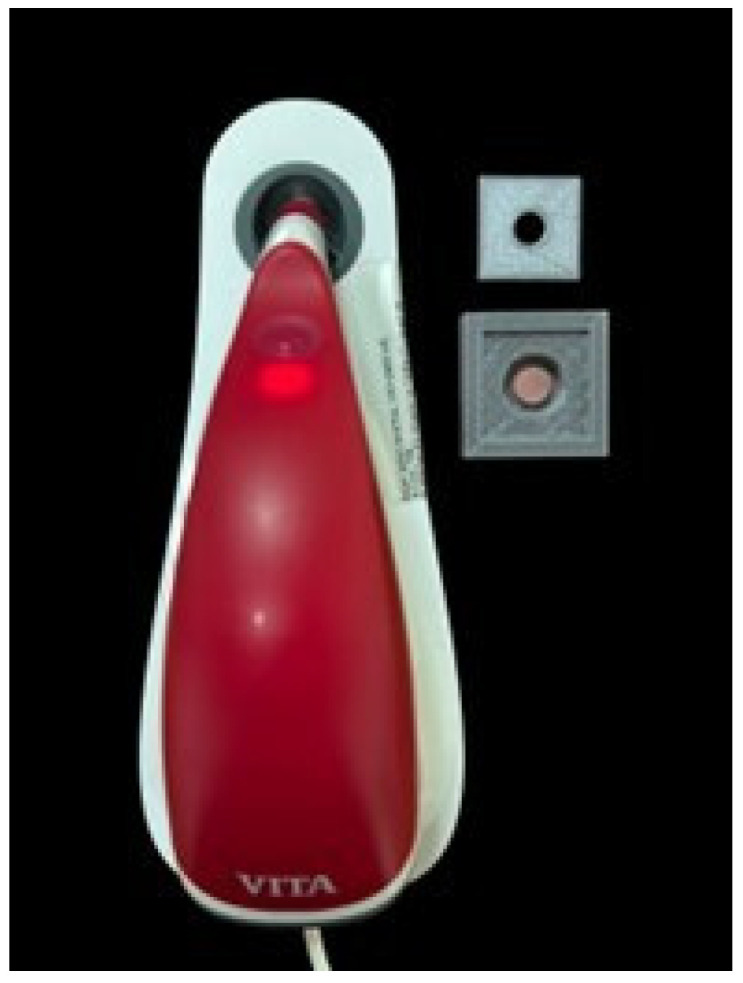
The experimental procedure for color evaluation.

**Figure 5 polymers-17-00056-f005:**
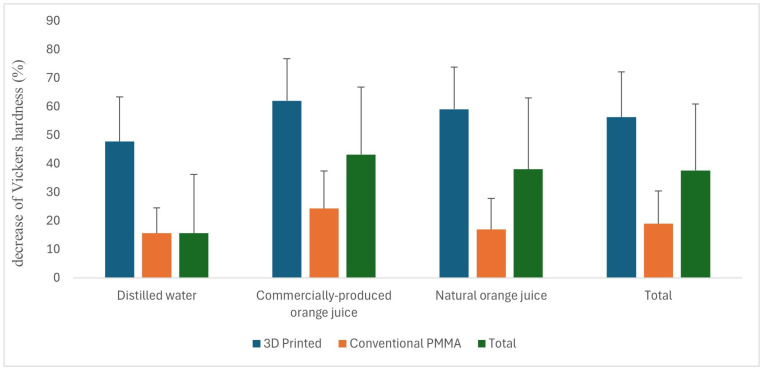
Mean values of the Vickers hardness based on material and solution.

**Figure 6 polymers-17-00056-f006:**
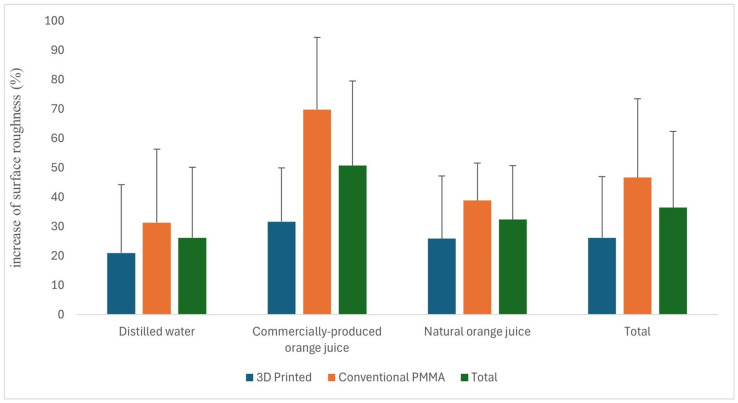
Mean values of the surface roughness based on material and solution.

**Figure 7 polymers-17-00056-f007:**
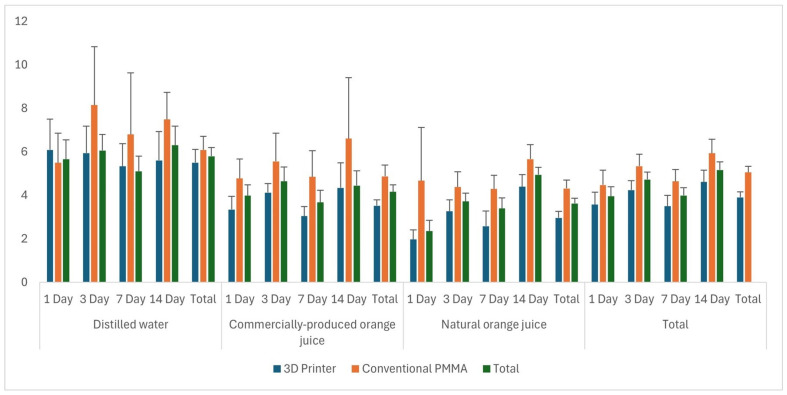
Mean ΔE values based on material, solution, and time.

**Table 1 polymers-17-00056-t001:** pH levels and composition of solutions.

Solution	pH	Composition
Distilled water	7.04	Water
Commercially produced orange juice	3.7	Water, sugar or fructose-glucose syrup, orange pulp (5%), orange juice concentrate, acidity regulators (citric acid, sodium citrate), thickening agents (gum arabic, glycerol esters of wood rosin), flavorings, antioxidant (ascorbic acid), colorant (beta-carotene).
Natural orange juice	5.3	%100 orange juice

**Table 2 polymers-17-00056-t002:** Comparison of the Vickers hardness (%) based on material and solution.

	Sum of Squares	Df	Mean Square	F	*p*	*η* ^2^
Material	20883.9	1	20883.9	119.292	<0.001	0.688
Solution	1323.068	2	661.534	3.779	0.029	0.123
Material × Solution	251.429	2	125.715	0.718	0.492	0.026

Df: Degrees of freedom; f: Analysis of variance test statistic; *η*^2^: Partial Eta Squared; R^2^: %70.38; Adj. R^2^: 67.63.

**Table 3 polymers-17-00056-t003:** Descriptive statistics for the Vickers hardness based on material and solution.

Solution		Material		Total
		3D-Printed	Conventional PMMA	
Distilled water	% decrease in Vickers hardness	47.71 ± 15.6	15.59 ± 8.92	31.65 ± 20.6 ^b^
	Vickers hardness baseline	45.69 ± 12.32	28.45 ± 3.54	37.07 ± 12.49
	Vickers hardness final	17.54 ± 4.15	23.9 ± 3.2	20.72 ± 4.86
Commercially produced orange juice	% decrease in Vickers hardness	61.98 ± 14.79	24.28 ± 13.16	43.13 ± 23.66 ^a^
	Vickers hardness baseline	46.12 ± 10.09	26.52 ± 4.87	36.32 ± 12.67
	Vickers hardness final	16.65 ± 4.33	19.61 ± 2.15	18.13 ± 3.66
Natural orange juice	% decrease in Vickers hardness	59.04 ± 14.74	16.92 ± 10.87	37.98 ± 25.02 ^ab^
	Vickers hardness baseline	42.6 ± 9.16	24.25 ± 4.49	33.43 ± 11.74
	Vickers hardness final	21.15 ± 3.98	19.85 ± 2.72	20.5 ± 3.38
Total	% decrease in Vickers hardness	56.24 ± 15.81 ^A^	18.93 ± 11.41 ^B^	37.59 ± 23.26
	Vickers hardness baseline	44.8 ± 10.36	26.41 ± 4.54	35.61 ± 12.2
	Vickers hardness final	18.45 ± 4.47	21.12 ± 3.3	19.78 ± 4.12

Mean ± standard deviation; final and baseline values are presented descriptively only and have not been statistically compared. ^a,b^: There is no significant difference between solutions with the same letter. ^A,B^ There is no significant difference between materials with the same letter.

**Table 4 polymers-17-00056-t004:** Comparison of the surface roughness based on material and solution.

	Sum of Squares	Df	Mean Square	F	*p*	*η* ^2^
Material	6309	1	6308.671	13.931	<0.001	0.205
Solution	6543	2	3271.311	7224	0.002	0.211
Material × Solution	2365	2	1182.401	2611	0.083	0.088

Df: Degrees of freedom; f: Analysis of variance test statistic; *η*^2^: Partial Eta Squared; R^2^: 38.36%; Adj. R^2^: 32.65%.

**Table 5 polymers-17-00056-t005:** Descriptive statistics for the surface roughness based on material and solution.

Solution		Material		Total
		3D-Printed	Conventional PMMA	
Distilled water	% increase in surface roughness	20.96 ± 23.21	31.3 ± 24.97	26.13 ± 24.06 ^b^
Commercially produced orange juice	% increase in surface roughness	31.63 ± 18.34	69.83 ± 24.57	50.73 ± 28.8 ^a^
Natural orange juice	% increase in surface roughness	25.87 ± 21.3	38.86 ± 12.69	32.36 ± 18.32 ^b^

^a,b^: There is no significant difference between solutions with the same letter.

**Table 6 polymers-17-00056-t006:** Comparison of ΔE values based on material, solution, and time.

	Test Statistics	*p* *
Material	9.173	0.004
Solution	14.253	0.002
Time	4.206	0.265
Material × Solution	0.149	0.930
Material × Time	0.207	0.978
Solution × Time	1.477	0.965
Material × Solution × Time	1.713	0.950

* Robust ANOVA; the trimmed mean method was used for comparison (with a trimming rate of 5%).

**Table 7 polymers-17-00056-t007:** Comparison of ΔE values based on material, solution, and time.

Solution	Time	Material		Total
		3D-Printed	Conventional PMMA	
Distilled water	1 Day	6.09 ± 1.41	5.49 ± 1.36	5.65 ± 0.89
	3 Day	5.94 ± 1.24	8.15 ± 2.68	6.06 ± 0.74
	7 Day	5.33 ± 1.04	6.8 ± 2.83	5.1 ± 0.69
	14 Day	5.6 ± 1.33	7.49 ± 1.24	6.3 ± 0.87
	Total	5.5 ± 0.6	6.09 ± 0.62	5.79 ± 0.41 ^b^
Commercially produced orange juice	1 Day	3.34 ± 0.61	4.77 ± 0.9	3.98 ± 0.5
	3 Day	4.12 ± 0.42	5.56 ± 1.29	4.64 ± 0.66
	7 Day	3.04 ± 0.44	4.85 ± 1.19	3.67 ± 0.56
	14 Day	4.34 ± 1.15	6.61 ± 2.8	4.44 ± 0.68
	Total	3.52 ± 0.26	4.87 ± 0.51	4.16 ± 0.31 ^a^
Natural orange juice	1 Day	1.97 ± 0.44	4.67 ± 2.45	2.35 ± 0.5
	3 Day	3.26 ± 0.52	4.38 ± 0.7	3.72 ± 0.37
	7 Day	2.58 ± 0.7	4.29 ± 0.62	3.39 ± 0.49
	14 Day	4.39 ± 0.55	5.66 ± 0.66	4.94 ± 0.34
	Total	2.96 ± 0.3	4.31 ± 0.38	3.62 ± 0.24 ^a^
Total	1 Day	3.57 ± 0.57	4.47 ± 0.68	3.95 ± 0.44
	3 Day	4.24 ± 0.43	5.34 ± 0.55	4.72 ± 0.35
	7 Day	3.5 ± 0.49	4.64 ± 0.54	3.99 ± 0.36
	14 Day	4.61 ± 0.54	5.93 ± 0.64	5.16 ± 0.37
	Total	3.9 ± 0.26 ^A^	5.05 ± 0.28 ^B^	

Trimmed mean ± standard error. ^a,b^: There is no significant difference between solutions with the same letter. ^A,B^: There is no significant difference between materials with the same letter.

## Data Availability

The original contributions presented in this study are included in the article. Further inquiries can be directed to the corresponding author.

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
