# Peer review of "Effect of Orange Juice on the Properties of Heat-Polymerized and 3D-Printed Denture Materials"

_polymers, 2024, doi:10.3390/polym17010056_

Round 1

Reviewer 1 Report

Comments and Suggestions for Authors

The study evaluates the color stability, surface roughness, and hardness of 3D 13 printed and conventional heat polymerized Polymethyl Methacrylate resin. Good experiments are conducted to compare the samples. However, the manuscript organization is and content should be improved. I hope the following comments help in this way:

1-     The title is a bit long and try to shorten it at least by removing the words “(PMMA)”, “base”, and “produced”, or use a different title.

2-     Abstract: needs severe modification. The sentences related to the experimental procedure in the abstract are mentioned with unnecessary details like: "were performed using a spectrophotometer 16 (VITA Easyshade® V, Germany)", and "using the Shapiro–Wilk, Tukey HSD", these details are normally written in the material and method section. On the other hand, we expect more detailed results in the abstract, while in the current version of paper it is not declared. for instance: "Surface roughness increase was lower in 3D-printed samples than". All the results are declared with “lower” and “higher” while we expect explicit quantitative results instead of general speaking.

3-     The last sentence in the abstract is redundant and will be indeed expressed by the quantitative results. Therefore delete: "The manufacturing method and interaction with various solutions significantly affected the color stability, surface roughness, and microhardness of the denture base materials.". That is a general and redundant sentence in the abstract.

4-     Introduction: the 3D printing method used to produce the samples should be mentioned and some works referenced.

5-     Experimental Procedure: at least a proper images should be provided showing the experimental setup, production process as well as the tests methods.

6-     What is the "(%)" used after "Vickers hardness" is it a dimension? unit?

7-     Could it be possible to remove clauses such as increase or decrease from the title of Tables? you know, when we say "Descriptive statistics for the increase in surface roughness (%) based on material " before looking to the results of table we actually concluded the increase.

8-     In Table 5, what does the Total mean? when the samples of 3d printed and PMMA are separate, or corrosive liquids are separate why they are combined and the average is reported?

9-     Fig 2: what is the ration behind the reporting the results in (%), while hardness and surface roughness have their own units.

Author Response

Reviewer 1

Comments: 1- The title is a bit long and try to shorten it at least by removing the words “(PMMA)”, “base”, and “produced”, or use a different title.

Response: In line with your advice, the title has been updated as follows; ‘Effect of Orange Juice on the Properties of Heat-Polymerized and 3D-Printed Denture Materials’

Comments: 2-     Abstract: needs severe modification. The sentences related to the experimental procedure in the abstract are mentioned with unnecessary details like: "were performed using a spectrophotometer 16 (VITA Easyshade® V, Germany)", and "using the Shapiro–Wilk, Tukey HSD", these details are normally written in the material and method section. On the other hand, we expect more detailed results in the abstract, while in the current version of paper it is not declared. for instance: "Surface roughness increase was lower in 3D-printed samples than". All the results are declared with “lower” and “higher” while we expect explicit quantitative results instead of general speaking.

Response: Thank you for your valuable feedback. In line with your advice, the relevant section has been revised as follows; ‘This study evaluated the color stability, surface roughness, and hardness of 3D-printed and heat-polymerized denture materials. A total of 90 samples were prepared, with equal numbers of 3D-printed and heat-polymerized discs. The initial hardness, surface roughness, and color values of the samples were measured. After 14 days of immersion in distilled water, natural orange juice, or commercial orange juice, the measurements were repeated. Based on the findings, 3D-printed samples exhibited a greater reduction in Vickers hardness (56.24 ±15.81%) compared to heat-polymerized samples (18.93 ±11.41%). Materials immersed in commercial orange juice exhibited a greater reduction in hardness compared to those in other solutions (43.13 ±23.66). Surface roughness increased by 46.66 ±26.8% in heat-polymerized samples and by 26.16 ±20.78% in 3D-printed samples, with the highest increase observed in commercial orange juice (50.73 ±28.8%) (p < 0.001). The color change (ΔE) was significantly higher in heat-polymerized samples (ΔE = 5.05 ±0.28) than in 3D-printed samples (ΔE = 3.9 ±0.26) (p < 0.001). This study demonstrates that the material type and immersion solutions play a critical role in determining the mechanical and optical properties of denture materials, with commercial orange juice having the most pronounced effect on surface roughness and hardness.

Comments: 3-     The last sentence in the abstract is redundant and will be indeed expressed by the quantitative results. Therefore delete: "The manufacturing method and interaction with various solutions significantly affected the color stability, surface roughness, and microhardness of the denture base materials." That is a general and redundant sentence in the abstract.

Response: Thank you for your valuable comment. In line with your suggestion, the last sentence of the summary section has been revised as follows; ‘This study demonstrates that the material type and immersion solutions play an important role in determining the mechanical and optical properties of denture materials, with commercial orange juice having the most pronounced effect on surface roughness and hardness.’

Comments: 4-     Introduction: the 3D printing method used to produce the samples should be mentioned and some works referenced.

Response: Thank you for your valuable feedback. In response to your comment, we have revised the introduction section to include information about the 3D printing method used in this study, specifically focusing on DLP technology. We have also added relevant references to support this section. The updated text now reads as follows:

“In dentistry, the most frequently preferred 3D printers are stereolithography (SLA) and digital light processing (DLP) models, depending on the applied method. In these devices, the printing platform is immersed in resin cured by ultraviolet (UV) or light-emitting diode (LED) light. The laser or LED light creates and shapes each layer of the object, repeating the process until the final print is completed [26–28]. While numerous studies have been conducted on CAD-CAM milled blocks [29–31], research on 3D-printed resins using DLP technology remains limited. With the increasing popularity of additive manufacturing techniques such as 3D printing, research in this field is becoming increasingly significant.”

We believe this revision addresses the comment effectively and provides the necessary context for the methodology and its relevance to the existing literature.

Comments: 5-     Experimental Procedure: at least a proper images should be provided showing the experimental setup, production process as well as the tests methods.

Response: Thank you for your valuable comments. In line with your comment, I have made the following additions to the article:

  1. Figures of the production process have been added as Figures 1a and 1b.
  2. A figure of the microhardness testing procedure has been added as Figure 2.
  3. A figure of the roughness testing procedure has been added as Figure 3.
  4. A figure of the color assessment procedure has been added as Figure 4.

These additions aim to provide a clear visual representation of both the manufacturing processes and the experimental methods.

Comments: 6-     What is the "(%)" used after "Vickers hardness" is it a dimension? unit?

Response: Thank you for pointing out this problem. The “(%)” after “Vickers hardness” refers to the percentage reduction from the initial hardness value. This is not a unit of measure or a dimension of Vickers hardness, but a relative percentage reduction calculated based on the initial and final measurements. For clarity, the percentage reduction is calculated using the following formula:

Percentage reduction = [(Initial hardness - Final hardness) / Initial hardness] × 100.

Please note that this clarification will be included in the revised article to avoid any confusion. Thank you for your consideration of this issue. Similar to the calculation of percentage decrease in Vickers hardness values, the percentage increase in surface roughness was calculated using the following formula:

Percentage increase = [(Final roughness - Initial roughness) / Initial roughness] × 100.

This calculation was performed to assess the relative increase in surface roughness after exposure to different solutions. This description has been added to the “Materials and Methods” section of the revised manuscript.

Comments: 7-     Could it be possible to remove clauses such as increase or decrease from the title of Tables? you know, when we say "Descriptive statistics for the increase in surface roughness (%) based on material " before looking to the results of table we actually concluded the increase.

Response: Thank you for your valuable comment. We have made the necessary adjustments to the expressions such as “increase” or “decrease” in the table and figure captions based on your comment. We have revised all table and figure captions to use neutral terminology, which is intended to ensure that the data is presented in an objective manner.

Comments: 8-     In Table 5, what does the Total mean? when the samples of 3d printed and PMMA are separate, or corrosive liquids are separate why they are combined and the average is reported?

Response: Thank you for your valuable comment. The “Total” column in Table 5 has been added to summarize the increase in surface roughness across different materials and solutions. However, we understand that combining these values and presenting an average may not be fully appropriate for analyzing the materials and solutions individually. In response to your comment, we have revised the table to remove the “Total” column and present the data based solely on the interactions of materials and solutions. We believe this will make the data more specific and free from misunderstandings. Thank you for raising this issue.

Comments: 9-     Fig 2: what is the ration behind the reporting the results in (%), while hardness and surface roughness have their own units.

Response: Thank you for your valuable comment. The results in Figure 2 are presented as percentage increase to provide a normalized comparison between the different materials and solutions. This approach was chosen because the initial surface roughness values vary between samples and allows for a more consistent interpretation of the data. Furthermore, our preference for this method is based on previous and published literature on this subject [1,2]. Using percentage changes instead of an analysis in original units makes it possible to compare different initial values. For the same reason, the results in Figure 1 are also presented as percentage reductions. Thank you again for your feedback and contributions.

Note: Figure numbers have changed after additions and revisions to the manuscript. The figure for mean values of the Vickers hardness became Figure 5. Mean values of the surface roughness became Figure 6.

  1. Rıbeıro, Juliana Silva, et al. In situ evaluation of color stability and hardness' decrease of resin‐based composites.Journal of esthetic and restorative dentistry, 2017, 29.5: 356-361.
  2. de Sousa Barbosa, Renata Pereira, et al. Effect of cariogenic biofilm challenge on the surface hardness of direct restorative materials in situ. Journal of Dentistry, 2012, 40.5: 359-363.

Reviewer 2 Report

Comments and Suggestions for Authors

Review Report on the Manuscript " polymers-3352910: Evaluation of Color Stability, Hardness, and Surface Roughness of Heat-Polymerized and 3D-Printed Polymethyl Methacrylate (PMMA) Denture Base Materials in Natural and Commercially Produced Orange Juice". The study employs robust experimental design, including the preparation of samples, systematic immersion in solutions, and detailed analysis using validated techniques such as spectrophotometry and Vickers hardness tests.  In addition, statistical analyses are well-presented, and the data effectively supports the conclusions drawn. Besides, the study provides valuable insights into the performance of 3D-printed versus heat-polymerized PMMA in real-world conditions, which is relevant to dental materials research and practice. Here are some suggestions to enhance the quality and clarity of the manuscript:

1.    While pH values and compositions were mentioned, their specific interactions with PMMA materials could be better explained, particularly the unexpected ΔE values in distilled water.

2.    Enhance clarity in Figure 3.

3.    Elaborate on how the findings align or diverge from existing literature on PMMA degradation in various media, especially the role of pH and ionic content in orange juice.

4.    Include a more detailed discussion on the underlying mechanisms affecting hardness, roughness, and color stability in 3D-printed versus conventional PMMA. For example, explore the role of material porosity or surface chemistry differences.

5.    Expand on the broader implications of the findings, such as their potential impact on clinical decisions for denture material selection.

6.    Simplify some of the text. For instance, avoid overuse of technical jargon without definitions. For example, “Polymers like PMMA may undergo hydrolytic degradation when they come into contact with water [50]. This process can lead to the breaking of polymer chains, thereby resulting in surface dullness or changes in color tone, which may contribute to the high ΔE values observed in samples immersed in distilled water." and the suggested revision:
[When PMMA interacts with water, it can absorb moisture, causing its structure to degrade. This degradation may break the material's polymer chains, resulting in surface dullness and noticeable changes in color."

7.    “14-day immersion” period may not fully represent long-term clinical exposure to such solutions!

8.    Address the potential variability introduced by using a single operator for the polishing process.

This study addresses an important topic in dental material science and presents significant findings regarding the comparative performance of heat-polymerized and 3D-printed PMMA. With additional insights into the mechanisms and refinement of presentation, the manuscript will make a valuable contribution to the field. Therefore, the submitted manuscript requires major revisions.

Author Response

Reviewer 2

Comments: 1. While pH values and compositions were mentioned, their specific interactions with PMMA materials could be better explained, particularly the unexpected ΔE values in distilled water.

Response: Thank you for drawing attention to this issue and allowing us to make additions to our article. In response to your criticism, we have expanded the description of the specific interactions of pH values and compositions with PMMA materials in our article. We have made the following additions; ‘Prolonged contact of resin-based materials with environments containing large amounts of water and acids can lead to expansion of the resin matrix, hydrolysis of the silane and consequently the formation of microcracks and voids [8]. In the literature, it has been reported that beverages containing acidity regulators such as phosphoric acid, malic acid and citric acid cause hydrolysis of ester radicals in PMMA monomers and disruption of the structural integrity of resin composite materials [51, 52]. Citric acid and sodium citrate in beverages such as commercial orange juice are examples of these effects. In particular, citric acid, due to its high calcium chelating capacity, can remove calcium from the saliva or prosthesis surface, making the resin surface more sensitive. Therefore, beverages with low pH and citric acid are known to have the highest corrosive effect on PMMA materials [53].

Our findings are in agreement with the study conducted by Al Taweel et al [43]. In that study, it was revealed that acidic beverages with low pH values such as Coca-Cola and lemon juice caused a significant increase in the surface roughness (Ra) of PMMA prosthetic resins. This effect was attributed to the fact that acidic components such as phosphoric acid and citric acid disrupt the structural integrity of the polymer surface, leading to dissolution and softening [54]. Again, Abu-Bakr et al. [55] revealed that low pH environments (such as orange juice and Coca-Cola) negatively affect the color stability and surface integrity of compomers. It was reported that under acidic conditions, the surface of the compomer softened significantly due to ion loss from the glass phase, resulting in a rougher structure with voids on the surface. The study showed that this could be attributed to the separation of individual particles [55].

Increased acidity in the oral environment can lead to many adverse conditions such as dental caries, periodontal disease, taste disturbance, bad breath and oral infections. Patients using dental prostheses may experience denture stomatitis as a result of fungal growth due to low pH. Furthermore, the effects of a decrease in oral pH on dental prosthetic acrylic resins are significant. Continuous contact with acidic saliva can lead to degradation of the acrylic material, a decrease in microhardness and the release of more residual monomers. This can lead to a decrease in the fracture resistance of the material and make prostheses more brittle [56, 57].

Similarly, Tinaztepe et al. [59] reported that higher roughness and hardness changes were observed in the groups immersed in low pH environment in their study investigating the hardness and surface roughness of prosthesis base acrylic resins produced by different methods.

In the literature, it has been reported that natural orange juice has higher sodium, potassium, calcium and copper content compared to commercial orange juice [60]. The higher mineral concentrations contained in natural orange juice are thought to cause less dissolution or degradation of the PMMA surface. In contrast, it is suggested that the lower mineral content and acidic modifiers contained in commercial orange juice may have led to a greater loss of surface hardness and more pronounced increases in surface roughness.

In response to your criticism, we have expanded the text as follows to better explain the specific interactions of PMMA materials with distilled water;

‘When PMMA interacts with water, it can absorb moisture, causing its structure to degrade [51]. This degradation may break the material's polymer chains, resulting in surface dullness and noticeable changes in color.

Some authors have reported that the water content of mouthwashes may affect discoloration; therefore, the effects seen in mouthwashes may also be observed in the distilled water (control) group [68, 69]. Ulusoy et al. also observed discoloration in all samples stored in distilled water in their study [70]. In accordance with these studies, color changes were observed in the distilled water groups in the present study.  Water absorption of resin materials is primarily affected by their chemical composition and the cavities they contain [71]. The chemical composition plays an important role in the chemical interaction with water molecules; this is particularly related to the hydrophilic polar regions in residual monomers, which tend to interact with water molecules [71, 72]. While the coloration effect of distilled water was expected to be the lowest among the groups, it is thought that the opposite result obtained may be due to the distillation method and device used [69].

We believe that these explanations contribute to a better understanding of the unexpected ΔE values in distilled water and their specific interactions with PMMA materials.

Comments: 2. Enhance clarity in Figure 3.

Response: Thank you for your valuable comment. In line with your comment about improving the clarity of Figure 3, necessary edits have been made to make not only this figure but all figures in the article clearer.

Comments: 3. Elaborate on how the findings align or diverge from existing literature on PMMA degradation in various media, especially the role of pH and ionic content in orange juice.

Response: Thank you for the opportunity to provide a more robust discussion on how our findings on the degradation of PMMA in different media, in particular the role of pH and ionic content in orange juice, agree or differ with the literature. In response to your comment, we have significantly expanded the relevant section in the paper.

We have discussed in detail the effects of low pH environments and supported this with relevant studies. For example, we discussed the effects of acidic components such as citric acid and sodium citrate present in commercial orange juice on PMMA surface properties. Studies discussing the negative effects of low pH beverages (e.g. Coca-Cola, lemon juice) on surface roughness and hardness and mechanisms such as hydrolytic degradation and calcium chelation are included. In addition, the differences between natural and commercial orange juices are also addressed, with particular emphasis on the relationship between the mineral content of these beverages (e.g., sodium, potassium, calcium and copper) and degradation on the PMMA surface. Consistent with our findings, it was argued that commercial orange juice caused more pronounced changes in surface roughness and hardness due to its lower mineral content and acidic modifiers, while natural orange juice showed relatively protective effects.

These additions not only strengthen the agreement of our study with the literature, but also highlight the unique contributions of this study. These points are clearly stated in the revised manuscript. The additions are in paragraphs 8, 10 and 11 of the discussion section and are as follows;

Prolonged contact of resin-based materials with environments containing large amounts of water and acids can lead to expansion of the resin matrix, hydrolysis of the silane and consequently the formation of microcracks and voids [8]. In the literature, it has been reported that beverages containing acidity regulators such as phosphoric acid, malic acid and citric acid cause hydrolysis of ester radicals in PMMA monomers and disruption of the structural integrity of resin composite materials [51, 52]. Citric acid and sodium citrate in beverages such as commercial orange juice are examples of these effects. In particular, citric acid, due to its high calcium chelating capacity, can remove calcium from the saliva or prosthesis surface, making the resin surface more sensitive. Therefore, beverages with low pH and citric acid are known to have the highest corrosive effect on PMMA materials [53].

Our findings are in agreement with the study conducted by Al Taweel et al. [43]. In that study, it was revealed that acidic beverages with low pH values such as Coca-Cola and lemon juice caused a significant increase in the surface roughness (Ra) of PMMA prosthetic resins. This effect was attributed to the fact that acidic components such as phosphoric acid and citric acid disrupt the structural integrity of the polymer surface, leading to dissolution and softening [54]. Again, Abu-Bakr et al. [55] revealed that low pH environments (such as orange juice and Coca-Cola) negatively affect the color stability and surface integrity of compomers. It was reported that under acidic conditions, the surface of the compomer softened significantly due to ion loss from the glass phase, resulting in a rougher structure with voids on the surface. The study showed that this could be attributed to the separation of individual particles [55].

Increased acidity in the oral environment can lead to many adverse conditions such as dental caries, periodontal disease, taste disturbance, bad breath and oral infections. Patients using dental prostheses may experience denture stomatitis as a result of fungal growth due to low pH. Furthermore, the effects of a decrease in oral pH on dental prosthetic acrylic resins are significant. Continuous contact with acidic saliva can lead to degradation of the acrylic material, a decrease in microhardness and the release of more residual monomers. This can lead to a decrease in the fracture resistance of the material and make prostheses more brittle [56, 57].

Similarly, Tinaztepe et al. [59] reported that higher roughness and hardness changes were observed in the groups immersed in low pH environment in their study investigating the hardness and surface roughness of prosthesis base acrylic resins produced by different methods.

In the literature, it has been reported that natural orange juice has higher sodium, potassium, calcium and copper content compared to commercial orange juice [60]. The higher mineral concentrations contained in natural orange juice are thought to cause less dissolution or degradation of the PMMA surface. In contrast, it is suggested that the lower mineral content and acidic modifiers contained in commercial orange juice may have led to a greater loss of surface hardness and more pronounced increases in surface roughness.

Comments: 4. Include a more detailed discussion on the underlying mechanisms affecting hardness, roughness, and color stability in 3D-printed versus conventional PMMA. For example, explore the role of material porosity or surface chemistry differences.

Response: Thank you for your valuable feedback on the need for a more detailed discussion on the mechanisms affecting hardness, roughness and color stability in 3D printed and conventional PMMA materials. Accordingly, we have revised the manuscript to comprehensively address these points. In particular, we have detailed the effects of double bond conversion, layer-by-layer production, curing time, material porosity, and surface chemistry differences on the results. We have utilized relevant literature to strengthen the discussion and have marked all changes made in the revised manuscript. We believe that these changes increase the scientific depth and clarity of our work. The additions are in paragraphs 4, 5 and 7 of the discussion section and are as follows;

The layer-by-layer manufacturing process of 3D printed materials and the lower double bond conversion rate can lead to insufficient curing of each layer. This can negatively affect the mechanical properties of the material by reducing chain crosslinking efficiency [39]. In agreement with our current findings, Prypic et al. [35] reported that 3D printed denture base materials had lower surface hardness values compared to other acrylic materials tested, including CAD/CAM and conventional heat-polymerized resins. The decrease in hardness can be attributed to factors such as the composition of the material, stress layers, thermal stress and water absorption [35].

In addition, it has been reported in the literature that the mechanical properties of PMMA prosthetic base materials are affected by factors such as polymerization method, degree of crosslinking and chemical composition [40]. Especially the curing time plays a critical role on the hardness values of 3D printed resins. Studies have shown that extending the curing time significantly increases the crosslinking rate (degree of conversion, DC), leading to higher hardness values [40]. In the present study, a curing time of 30 minutes was used in accordance with the manufacturer's instructions. It is thought that if the curing time is extended, the hardness loss in 3D printed samples may be less and the results may differ. However, although the 3D printing process offers the advantage of fast production, the layer-by-layer manufacturing mechanism may lead to insufficient curing of each layer. This can lead to insufficient curing density throughout the structure and a significant reduction in chain crosslinking efficiency [41]. These limitations in the printing process may explain the observed differences in mechanical properties between 3D printed and conventional denture base materials.

The traditional prosthetic base fabrication method has been widely used for many years; however, problems such as porosity, surface roughness, and volumetric/linear shrinkage have been frequently reported [47, 48]. It has also been reported that rough surfaces retain more microorganisms than smooth surfaces and create a protected environment that is difficult to remove by oral hygiene methods [49, 50].

Comments: 5. Expand on the broader implications of the findings, such as their potential impact on clinical decisions for denture material selection.

Response: Thank you for your suggestion to discuss the broader implications of the findings of our study and their potential impact on clinical decisions for prosthetic material selection. Accordingly, we have addressed the potential advantages and limitations of these materials in different clinical scenarios, highlighting the differences in hardness, roughness, and color stability between 3D-printed and conventional PMMA materials in the manuscript. We have also detailed the effects of natural and commercial orange juices on material properties and related our findings to the effects of dietary habits and prosthetic material durability. These revisions aim to discuss the clinical relevance of our study more comprehensively and all changes are marked in the revised manuscript.

The additions were made to the 2 paragraphs of the discussion section before limitations and read as follows; 'These findings highlight important considerations for clinical decisions in the choice of prosthetic base material. The higher loss of microhardness observed with 3D printed materials may warrant caution in patients with high consumption of acidic beverages. However, the lower surface roughness and discoloration of these materials make them a more preferable option for patients who prioritize aesthetics and oral hygiene. Conventional PMMA materials, despite exhibiting higher hardness stability, may require more frequent maintenance due to their higher surface roughness and tendency to stain retention. Our findings also reveal that commercial orange juice shows more negative effects on material properties compared to natural orange juice. Commercial orange juice caused greater loss of hardness and more pronounced increases in surface roughness. This supports the hypothesis that the acidic modifiers and lower mineral content in commercial orange juice accelerate degradation of the PMMA surface. In contrast, the higher mineral concentrations of natural orange juice may have shown a relatively protective effect, leading to less surface dissolution and degradation.

These results emphasize the need to consider not only the choice of material but also the dietary habits of prosthesis wearers. In particular, regular consumption of commercial acidic beverages may adversely affect the mechanical and aesthetic properties of prosthetic materials, and it is of great importance for clinicians to provide appropriate care recommendations to patients. Material selection should be tailored to the individual needs and lifestyles of patients, balancing durability with aesthetic and functional requirements.'

Comments: 6. Simplify some of the text. For instance, avoid overuse of technical jargon without definitions. For example, “Polymers like PMMA may undergo hydrolytic degradation when they come into contact with water [50]. This process can lead to the breaking of polymer chains, thereby resulting in surface dullness or changes in color tone, which may contribute to the high ΔE values observed in samples immersed in distilled water." and the suggested revision:
[When PMMA interacts with water, it can absorb moisture, causing its structure to degrade. This degradation may break the material's polymer chains, resulting in surface dullness and noticeable changes in color."

Response: Thank you for your valuable feedback, which provided us with the opportunity to improve the clarity of the manuscript. In line with your suggestion, the original text:
‘Polymers like PMMA may undergo hydrolytic degradation when they come into contact with water [50]. This process can lead to the breaking of polymer chains, thereby resulting in surface dullness or changes in color tone, which may contribute to the high ΔE values observed in samples immersed in distilled water.’ has been simplified to: ‘When PMMA interacts with water, it can absorb moisture, causing its structure to degrade. This degradation may break the material's polymer chains, resulting in surface dullness and noticeable changes in color.

Comments: 7. “14-day immersion” period may not fully represent long-term clinical exposure to such solutions!

Response: Thank you for your valuable feedback that the 14-day immersion period is not fully representative of long-term clinical exposure. This duration is a standard approach commonly used in similar in vitro studies to evaluate material properties under controlled conditions. The immersion times used in our study were chosen based on protocols used in similar studies [1,2,3,4]. In Güler et al.'s study [5], a 24-hour immersion period simulated 1 month of coffee consumption, whereas in our study, a 14-day immersion period was chosen as the maximum simulated exposure period and represented a period of approximately 14 months under real-life conditions. This duration offers the possibility to comprehensively assess the short- and medium-term effects of the tested solutions on the materials. We have included this explanation in our paper to make the rationale for this choice clearer.

However, we recognize that longer immersion times may be more representative of long-term clinical conditions. The current study provides valuable information on the early and mid-term effects of these solutions. We think that future studies involving longer immersion times or accelerated aging protocols would be useful for long-term evaluation of material performance. We have added this limitation to our paper. The added part was as follows; ‘Furthermore, the immersion durations used in this study were limited to simulate short- to mid-term exposures. Longer immersion periods are needed to better represent long-term clinical conditions and provide a more comprehensive understanding of the tested solutions' effects on denture base materials.’

  1. Tantanuch, Saijai, et al. Surface roughness and erosion of nanohybrid and nanofilled resin composites after immersion in red and white wine. Journal of Conservative Dentistry and Endodontics, 2016, 19.1: 51-55.
  2. Tanthanuch, Saijai, et al. In vitro surface and color changes of tooth-colored restorative materials after sport and energy drink cyclic immersions. BMC Oral Health, 2022, 22.1: 578.
  3. Wayakanon, Praween; Narakaew, Teeraphan; Wayakanon, Kornchanok. Effects of various beverages on characteristics of provisional restoration materials. Clinical and Experimental Dental Research, 2024, 10.2: e842.
  4. Alencar-Sılva, Flávia J., et al. Effect of beverage solutions and toothbrushing on the surface roughness, microhardness, and color stainability of a vitreous CAD-CAM lithium disilicate ceramic. The Journal of prosthetic dentistry, 2019, 121.4: 711. e1-711. e6.
  5. Guler, Ahmet Umut, et al. Effects of different drinks on stainability of resin composite provisional restorative materials. The Journal of prosthetic dentistry, 2005, 94.2: 118-124.

Comments: 8.    Address the potential variability introduced by using a single operator for the polishing process.

Response: Thank you for your feedback that polishing by a single person can create variability. In this study, the polishing procedure was performed by one of the study authors, B. T., a prosthodontist with extensive clinical experience. This was chosen to ensure consistency of the procedure and to follow a standardized protocol. The protocol included three polishing stages (coarse, medium and fine grit), each performed at a speed of 300 rpm for 15 seconds. While the use of a single operator reduces inter-operator variability, we recognize that operator-dependent variability is not completely eliminated. To address this limitation, all polishing steps were performed under controlled conditions using consistent pressure, time and polishing wheels.

Reviewer 3 Report

Comments and Suggestions for Authors

The paper presented a nature orange juice additive to improve the color stability, hardness and surface roughness of the 3D printed PMMA Denture base material. The paper is lack of scientific data and explanations for the improvement of PMMA polymer chemical and physical properties in additive 3D printing. It is not acceptable to be publish in polymer journal, and I suggest to publish in journals for denture application journals.

Author Response

Reviewer 3

Comments: The paper presented a nature orange juice additive to improve the color stability, hardness and surface roughness of the 3D printed PMMA Denture base material. The paper is lack of scientific data and explanations for the improvement of PMMA polymer chemical and physical properties in additive 3D printing. It is not acceptable to be publish in polymer journal, and I suggest to publish in journals for denture application journals.

Response: Thank you for your valuable evaluation and feedback regarding our article. However, we have noticed that the assessment misunderstood the main objective of our study. We did not aim to add any contributions to 3D printed PMMA prosthetic base materials. Instead, our goal was to investigate the effects of orange juice, a commonly consumed beverage with a low pH, on 3D printed and conventional PMMA materials. Specifically, to explore changes in mechanical and surface properties such as surface roughness, hardness, and color stability, our samples were immersed in natural and commercial orange juice for a period of 14 days.

In this context, we believe that understanding the effects of widely consumed beverages like orange juice on dental prosthetic materials provides important insights regarding the longevity of prostheses and patient satisfaction. Our article thoroughly discusses the literature concerning the effects of low pH on PMMA materials in terms of mechanical, surface, and chemical changes, and our findings are compared with similar studies in the literature. Additionally, we emphasize the differing effects between commercial and natural orange juice, highlighting the role of these beverages' mineral content and acidic regulators on the properties of the materials.

Taking your critiques into account, we have made significant additions to enhance the scientific rigor and comprehensiveness of the text. These additions are based on the literature discussing the effects of low pH on PMMA materials regarding mechanical, surface, and chemical changes. Moreover, our findings have been contextualized more robustly by comparing them with similar studies in the literature. We believe these improvements have clarified the scope and objective of the study and have contributed to a better understanding of the chemical and physical mechanisms affecting PMMA.

To clarify the misunderstanding, we would like to mention that we will include an explanation in our article that we believe should be taken into account. We are confident that our findings provide valuable contributions from the perspective of polymer chemistry, and we trust that this work is suitable for the target audience of the polymer journal. We also acknowledge that the results of this study carry significant implications for dental prosthetic applications and could serve as a useful reference for readers in these fields.

We appreciate the opportunity your evaluation has provided for better understanding and improving our article, and we thank you for your contributions.

Round 2

Reviewer 1 Report

Comments and Suggestions for Authors

No more comments

Reviewer 2 Report

Comments and Suggestions for Authors

It could be accepted.